# Operationalizing Quantized Disentanglement

## Abstract

Recent theoretical work established the unsupervised identifiability of quantized factors under any diffeomorphism. The theory assumes that quantization thresholds correspond to axis-aligned discontinuities in the probability density of the latent factors. By constraining a learned map to have a density with axis-aligned discontinuities, we can recover the quantization of the factors. However, translating this high-level principle into an effective practical criterion remains challenging, especially under nonlinear maps. Here, we develop a criterion for unsupervised disentanglement by encouraging axis-aligned discontinuities. Discontinuities manifest as sharp changes in the estimated density of factors and form what we call *cliffs*. Following the definition of independent discontinuities from the theory, we encourage the location of the cliffs along a factor to be independent of the values of the other factors. We show that our method, *Cliff*, outperforms the baselines on all disentanglement benchmarks, demonstrating its effectiveness in unsupervised disentanglement.

## 1 Introduction

Representation learning aims to find useful inductive biases that reflect the nature and structure of the data. In essence, the representation is desired to be disentangled, having modular factors of variation that control or cause the observed variables (Bengio et al., 2013; Eastwood et al., 2023). This is more precisely defined through identifiability theory, which provides mathematical conditions to determine when such factors can be uniquely recovered from observations. However, this problem remains hard to solve and difficult to apply to real-world data; some reasons for this discrepancy could be because of strict underlying assumptions from identifiability theory, or because the methods were designed in small and controlled settings that do not scale well. As an alternative to complete disentanglement, we use quantization as an inductive bias. Quantized latent factors are often a natural representation for humans, for example, when thinking of colors (that are continuous in the sensorial reality, but discrete when thinking of *red* or *blue*) and concepts **?**. We incorporate this inductive bias since it is a relaxed, yet useful form of disentanglement.

Disentanglement aims to find the axis where the ground truth latent factors lie. In this work, we develop the idea of axis alignment by leveraging discontinuities in the density of the latent factors. Inspired by the theoretical results in the literature concerning the identifiability of quantized latent factors (Barin-Pacela et al., 2024), we design a learning criterion to align with the axes the discontinuities in the learned latent density.

The theory assumes that these discontinuities are aligned with the axes in the joint probability density of the latent factors; equivalently, these are defined as *independent discontinuities* (formal definition in Appendix B.1). That allows the recovery of the axis alignment even after the warping of the latent variables by a nonlinear transformation. Hence, the theory uses these axis-aligned discontinuities as quantization thresholds.

We address the main limitations found in the empirical implementation of the criterion from Barin-Pacela et al. (2024), which was based on estimating gradients in the joint of the density of reconstructed factors and aligning them with the axis. We leverage the definition of independent discontinuities through conditionals, such that the location of the cliffs along a factor is independent of the values of the other factors. This criterion is combined with the encouragement of cliffs in the marginals and a term that avoids degenerate solutions.

Our approach makes this criterion suitable for nonlinear transformations and applies it to disentanglement datasets, which were unexplored in previous research.

Therefore, the contributions of this body of work are:

- The proposal of a new criterion that encourages axis alignment, and validation of its effectiveness on synthetic datasets under nonlinear transformations. This criterion is model-agnostic and can be applied as a regularizer to any model encoding a representation.

- Benchmarking and evaluation of the criterion and baselines on disentanglement datasets.

## 2   A brief overview of quantized identifiability

We learn from observed variables $x = (x_1, \ldots, x_D) \in \mathcal{X}$. The generative process assumes unobserved latent factors $z = (z_1, \ldots, z_d) \in \mathcal{Z}$ that are transformed into observed variables through a *mixing function* $f : \mathcal{Z} \to \mathcal{X}$, such that $x = f(z)$. We learn a function (encoder) $g : \mathcal{X} \to \mathcal{Z}$ that should approximate $f^{-1}$.

The field of causal representation learning studies identifiability by specifying conditions and assumptions on the generative model to reconstruct $z$ and $g$ up to certain indeterminacies, in the best case up to scaling and permutation. This setting is typically studied through independent component analysis, where the latent factors $z$ are assumed to be statistically independent. It has been long-established that when $f$ is nonlinear, independence alone can be easily satisfied and not enough to determine a unique factorization of the factors (Hyvärinen & Pajunen, 1999), deeming the unsupervised identifiability of latent factors impossible under a diffeomorphic map (Locatello et al., 2019; Buchholz et al., 2022) in the absence of either a stronger inductive bias on the map, weak supervision, or auxiliary information. This formulation allows for a technical definition of disentanglement through identifiability theory: reconstructing the true factors $z$ from observed data up to the indeterminacies.

Given the difficulty of *precise* unsupervised identifiability, Barin-Pacela et al. (2024) established the identifiability of a *quantization* of continuous factors under any diffeomorphic map. This theoretical result does not require independent factors but assumes the presence of *independent discontinuities* in their joint probability density function (pdf).

By definition, an axis-aligned discontinuity is equivalent to an independent discontinuity. A factor $z_i$ is said to have an independent discontinuity at $z_i = \tau$ if the joint pdf $p(z_1, \ldots, z_d)$ is discontinuous at $z_i = \tau$ regardless of the values taken by the other factors. A set of such independent discontinuities, for all factors, forms an axis-aligned grid. Importantly, the location of each factor's discontinuities can be used as a quantization threshold to yield its quantized value $q_i(z_i)$. When these latent factors $z$ are mapped to an observation $x$ by any diffeomorphism, the discontinuities are preserved in the pdf of $x$. They may no longer be axis-aligned, but the discontinuity grid structure survives under potential warping. It is, thus, possible to learn a reverse diffeomorphism that realigns discontinuities with the axes. This suffices to obtain recovered factors $(z'_1, \ldots, z'_d)$, together with new quantization thresholds, based on their discontinuities locations. Barin-Pacela et al. (2024) formally proved that their quantized values are, then, guaranteed to match the quantized values of the original factors, up to permutation and axis reversal. The details of the theorems are available in Appendix B and a discussion on the differences between Cliff and their preliminary criterion is available in Appendix B.3.

## 3   Related work

Existing literature has explored the quantization of latent factors in theory and practice. Kong et al. (2024) provide theoretical guarantees for learning discrete concepts from high-dimensional data through a hierarchical causal model, expanding upon the identifiability theory of discrete auxiliary variables from Kivva et al. (2022). However, realistic methods based on theoretical principles are still to be shown. Different kinds of quantization have been successfully proposed, such as vector quantization, in the case of the VQ-VAE (van den Oord et al., 2017), and Finite Scale Quantization (FSQ) (Mentzer et al., 2024). FSQ is a factorized quantization which fixes the binning and tries to fit the representation that preserves the information when

| Method | correlated latents | density constraint | general diffeomorphism | no auxiliary info | identifiability |
|---|---|---|---|---|---|
| $\beta$-VAE (Higgins et al., 2017) | ✓ | global | ✓ | ✓ | ✗ |
| HFS (Roth et al., 2023) | ✓ | global | ✓ | ✓ | ✗ |
| IOSS (Wang & Jordan, 2021) | ✓ | global | ✗ | ✓ | ✓ |
| (Kong et al., 2024) | ✓ | global | ✓ | ✓ | ✓ (quantized) |
| Additive Decoders (Lachapelle et al., 2023) | ✓ | non-global | ✗ | ✓ | ✓ (block) |
| Cliff (Barin-Pacela et al., 2024, and here) | ✓ | non-global | ✓ | ✓ | ✓ (quantized) |

Table 1: Contrasting theoretical guarantees of unsupervised disentanglement approaches. Cliff (following from Barin-Pacela et al. (2024)) is the only approach with minimal assumptions (no auxiliary info, no global density constraints, allows for correlated latent variables and general diffeomorphisms) that still allows for (quantized) identifiability guarantees.

quantized into this fixed binning. In contrast, our method aims to learn a natural quantization of factors based on properties that are preserved under a diffeomorphism, which makes it theoretically sound. One recent direction proposed by Hsu et al. (2024) builds on quantizing latent variables (Hsu et al., 2023; Mentzer et al., 2024) and combines it with three other inductive biases for disentanglement: encoding into independent latent variables (Chen et al., 2018) and having these variables interact minimally to generate data (Peebles et al., 2020). In contrast, this work is motivated to allow correlations between the latent variables.

Compared to other disentanglement methods that allow for correlations between the latent variables (Roth et al., 2023; Träuble et al., 2021; Wang & Jordan, 2021; Morioka & Hyvärinen, 2024), our work is more general as it explores the idea of latent quantization, hence requiring fewer and weaker assumptions. In particular, the theoretical results (Barin-Pacela et al., 2024) do not require factorized support (Roth et al., 2023; Wang & Jordan, 2021; Ahuja et al., 2022b) or knowledge about the grouping structure of observed variables (Morioka & Hyvärinen, 2024).

Table 1 compares the main methods for unsupervised disentanglement. We distinguish between *global* and *non-global* constraints on the density of latent factors $p_z$, with the purpose of judging the strength of the assumptions and their usefulness in practice. Hence, here "global" refers to when the main assumption on $p_z$ cannot be checked by simply looking at the neighborhood of points (for instance, factorized support requires global alignment between the boundaries). Alternatively, "non-global" means that the main assumption on the density can be checked by only looking locally at the distribution, using only a small neighbourhood around each point (for instance, a cliff). Furthermore, we clarify that while Wang & Jordan (2021) provides an identifiability proof, it makes a very strong assumption on the mixing map that is not enforced in the proposed method and has been replaced by an assumption on the map or auxiliary (interventional) information in follow-up work (Ahuja et al., 2022b). Finally, the table illustrates our proposal to develop an empirical method with identifiability guarantees that has fewer and weaker assumptions than existing methods.

## 4 A new regularizer: Cliff Alignment (Cliff)

The disentanglement principle suggested by the quantized identifiability theory (Barin-Pacela et al., 2024) is simple to state: "Learn a diffeomorphism that maps observations to recovered factors such that the discontinuities in the joint pdf of these factors form an axis-aligned grid". But translating this high-level principle into an effective practical criterion is far from straightforward. We wish to design an experimental criterion that encourages the model to learn discontinuities in the latent space, and for them to be independent (aligned with the axes).

*To lighten the notation, for the remainder of the paper, we will refer to the estimated latent factor $z_i'$ as $z_i$.*

We consider a finite sample of $n$ observed data points $\mathcal{D} = \{x^{(1)}, \ldots, x^{(n)}\}$ that are mapped to their recovered representations $\{z^{(1)}, \ldots, z^{(n)}\}$ via a learned parameterized function (such as a neural network) $\phi_\theta$, i.e. $z^{(k)} = \phi_\theta(x^{(k)})$ where $z^{(k)}$ denotes the vector of all factors of the $k$-th training point. We use kernel density

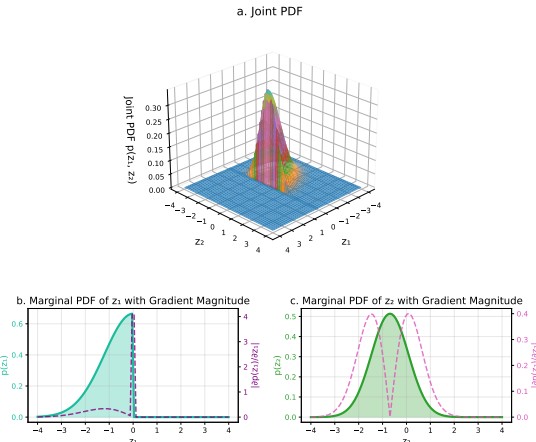

Figure 1: **a.** Joint PDF $p(z_1, z_2)$ with a *cliff* at $z_1 = 0$, parallel (aligned) to the $z_2$ axis. **b.** Marginal PDF of $z_1$, $p(z_1)$, displayed on the left axis; magnitude of the gradient $\partial p(z_1)/\partial z_1$ on the right axis (on a different scale). The magnitude of the gradient is high at the cliff, which is a point of sharp change in the marginal density. **c.** Marginal PDF of $z_2$, $p(z_2)$, and its repective gradient magnitude; no cliffs observed along this axis.

estimation to approximate the true probability density of recovered factors $p(z)$ in the finite-sample setting. We employ a Gaussian kernel of fixed bandwidth $\sigma$ (Parzen Windows estimator).

$$\hat{p}_\sigma(z) = \frac{1}{n}\sum_{k=1}^{n}\mathcal{N}(z; z^{(k)}, \sigma^2 I), \tag{1}$$

where $\mathcal{N}(z; z^{(k)}, \sigma^2 I)$ denotes the evaluation at $z$ of the pdf of a Gaussian of mean $z^{(k)}$ and diagonal covariance $\sigma^2 I$.

This yields a kernel-smoothed version of the true density. In this smoothed version, discontinuities are also smoothed, so that they will appear as cliffs with a steep slope (large but finite derivative) instead of actual discontinuities (infinite derivative). Therefore, we aim to design a criterion that encourages the pdf to have cliffs of high slope aligned with the axes.

One additional practical difficulty is that, while kernel density estimation works well in low dimensions, in high dimensions it tends to be quite challenging and unreliable. Luckily, the characteristic of independent (axis-aligned) discontinuities in the joint pdf $p(z_1, \ldots, z_d)$ should also be present in subsets of factors, as illustrated in Figure 1. Therefore, in practice, we relax our characterization to encourage a mapping that yields $z$ with:

1. discontinuities in marginal pdfs $p(z_i)$ *(univariate criterion)*.

2. discontinuities that are independent, in all pairs of factors $p(z_i, z_j)$, with $j \neq i$ *(bivariate criterion)*.

Hence, we can simply use low-dimensional kernel density estimates to estimate $p(z_i)$ and $p(z_i, z_j)$.

As motivated above, we can consider the presence of derivatives of high magnitude in the estimated density as evidence for the presence of discontinuities in the true pdf that has been kernel-smoothed. Note that the scale of the factors yielded by the mapping $\phi_\theta$ is irrelevant for the characterization of their pdf as having independent discontinuities. Thus, we first standardize each factor $z_i$ to have zero mean and unit variance, and then use a fixed-width kernel density estimation.

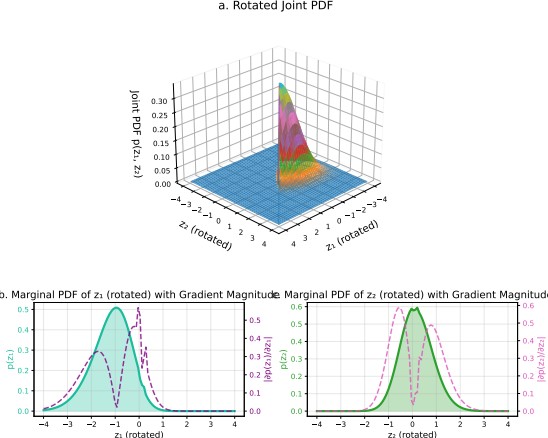

Figure 2: **a.** Joint PDF with $p(z_1, z_2)$ with a cliff parallel to the diagonal, therefore not axis-aligned – this is the same density as in Figure 1, but rotated by 45°. **b.** Marginal PDF of $z_1$, $p(z_1)$ and its respective gradient magnitude $\partial p(z_1)/\partial z_1$. **c.** Marginal PDF of $z_2$, $p(z_2)$, and its respective gradient magnitude. No cliffs observed in the marginals because this cliff is not axis-aligned. Note that while there are some bumps in the gradient magnitude, the scale of this magnitude is small, especially when compared to the cliff from Figure 1.

*To simplify notation, from here on, $p(z_i)$ and $p(z_i, z_j)$ will no longer denote the true pdf of the original $z$, but the result of the 1d or 2d kernel density estimate $\hat{p}_\sigma$ on standardized $z_i$ or $(z_i, z_j)$.*

Specifically, for each batch of size $n$, we first standardize $z$, and then compute:

$$p(z_i) = \frac{1}{n} \sum_{k=1}^{n} \mathcal{N}(z_i; z_i^{(k)}, \sigma^2) \tag{2}$$

and

$$p(z_i, z_j) = \frac{1}{n} \sum_{k=1}^{n} \mathcal{N}((z_i, z_j); (z_i^{(k)}, z_j^{(k)}), \sigma^2 I). \tag{3}$$

Furthermore, our criterion is based primarily on (partial) *derivatives* of the estimated densities, which can be computed in a similar way:

$$\frac{dp(z_i)}{dz_i} = \frac{1}{n} \sum_{k=1}^{n} \frac{d}{dz_i} \mathcal{N}(z_i; z_i^{(k)}, \sigma^2) \tag{4}$$

and

$$\frac{\partial p(z_i, z_j)}{\partial z_i} = \frac{1}{n} \sum_{k=1}^{n} \frac{\partial}{\partial z_i} \mathcal{N}((z_i, z_j); (z_i^{(k)}, z_j^{(k)}), \sigma^2 I), \tag{5}$$

where the (partial) derivative of the Gaussian kernel has a straightforward analytic expression.

The remainder of the section defines each term of the training objective.

### 4.1 Univariate criterion – Encouraging cliffs in the marginals

This term of the criterion aims to encourage the curve corresponding to the magnitude of the gradients of the marginal $\left| \dfrac{dp(z_i)}{dz_i} \right|$ to be very peaky, to have a few steep spikes (Figure 1). When this magnitude is high

enough, we call it a **cliff**. From the perspective of information theory, we would like the distribution to be as far as possible from a uniform distribution, which is the distribution with finite support that maximizes the entropy. This is a similar approach taken by independent component analysis (ICA), which attempts to move away from Gaussian distributions by maximizing the negentropy, since for infinite support, the Gaussian is the distribution that maximizes the entropy with specified mean and variance. However, here we are focusing not on the density, but on its derivative $\frac{dp(z_i)}{dz_i}$. Therefore, we formulate the following term for the criterion. We define $s_i$ to be the magnitude of the derivative of the marginal density of a factor

$$s_i(z_i) = \frac{1}{c}\left|\frac{dp(z_i)}{dz_i}\right|,\tag{6}$$

where $c$ is the normalization constant that ensures $s$ integrates to 1:

$$c = \int_{\mathcal{Z}_i}\left|\frac{dp(z_i)}{dz_i}\right| dz_i.\tag{7}$$

We can interpret $s_i$ as a pdf which is high wherever the derivative of $p(z_i)$ has a high magnitude. Then, we compute its differential entropy:

$$H(s_i) = -\int s_i(z_i)\log(s_i(z_i))\, dz_i\tag{8}$$

The entropy is a measure of "peakiness": it is the lowest when high magnitude derivatives are concentrated around one or a few points, which is what we want to encourage with this term.

Therefore, the univariate criterion hereby proposed simply adds the entropy for the density derivative of each factor:

$$l_{\mathrm{uni}} = \sum_{i=1}^{d} H(s_i).\tag{9}$$

Since the goal is to minimize the entropy, $l_{\mathrm{uni}}$ should be minimized.

Intuitively, this criterion not only encourages a spiky landscape but also aligns the spikes with the axes. As an illustration, Figure 1.b plots the ground-truth gradient magnitude and its respective marginal density. While there could be more distributions that result in the gradient function shown, the goal is to recover the underlying marginal (green), which has axis-aligned cliffs. This argument is validated in Figure 5, where it is visible that this criterion is optimal at the axis-aligned directions.

Second, we elaborate on how axis-aligned cliffs can be detected from the marginals. Suppose that there is a discontinuity in the joint $p(z_i, z_j)$ that is not axis-aligned, then the discontinuity will not be steep enough in the marginal, as illustrated in Figure 2. Since the marginal cliffs are sharper and more pronounced than the joint cliffs, the univariate criterion will encourage the marginals to have cliffs that are aligned with the axes.

As a technical remark, note that the Dirac delta distribution could be a solution for equation 9, and this solution will be avoided by counterbalancing it with another term in the loss function, as will be explained in section 4.3. The criterion we develop involves several one-dimensional integrals along a factor, used above to define $c$ or $H(s_i)$. In practice, they are estimated via basic numerical integration:

$$\int f(z_i)\, dz_i \approx \frac{b-a}{K}\sum_{z_i \in \mathcal{I}(a,b,K)} f(z_i),\tag{10}$$

where $\mathcal{I}(a,b,K)$ is a set of $K$ equally-spaced values between $a$ and $b$.[1]

---

[1] In practice, to estimate integrals along our standardized $z_i$, we use $\mathcal{I}(-5, +5, 100)$.

## 4.2 Bivariate criterion – Encouraging independent cliffs

For a visual understanding of independent discontinuities, Figure 1 illustrates an independent cliff, while Figure 2 illustrates a cliff that is not independent, as it is not aligned with the axes. Note that the cliff in Figure 1.a is independent since, for any value of $z_2$, there is always a discontinuity at $z_1$, therefore being independent of the value of $z_2$ (hence in agreement with the definition B.1). On the other hand, the cliff in Figure 2.a is not independent, as there is not a particular value of $z_1$ for which there would be a discontinuity at any value of $z_2$ or vice-versa.

To encourage the discontinuities to be independent, we look at different assignments of $z_j$ in $\dfrac{\partial p(z_i|z_j)}{\partial z_i}$, and expect all of these different functions to be similar or "close to each other". In particular, we look at the location of the peaks of these functions, which are the $z_i$ that lead to high-magnitude $\left|\dfrac{\partial p(z_i|z_j)}{\partial z_i}\right|$, and the location of these peaks should be the same across all the different assignments of $z_j$.

First, we deduce the derivative of the conditional $\dfrac{\partial p(z_i|z_j)}{\partial z_i}$ from the kernel density estimates in Eq. 2 and Eq. 5 as follows:

$$\frac{\partial p(z_i|z_j)}{\partial z_i} = \frac{1}{p(z_j)}\frac{\partial p(z_i, z_j)}{\partial z_i}. \tag{11}$$

Then, we define "density derivative magnitudes" as

$$u_{ij}(z_i|z_j) = \left|\frac{\partial p(z_i|z_j)}{\partial z_i}\right|, \tag{12}$$

and the corresponding normalized density as

$$\tilde{p}_{ij}(z_i|z_j) = \frac{u_{ij}(z_i|z_j)}{\int u_{ij}(z_i'|z_j)dz_i'}. \tag{13}$$

Here, $\tilde{p}_{ij}(\cdot|z_j)$ can be interpreted as a new probability density function, which is concentrated wherever the derivative of $p(z_i|z_j)$ has a high magnitude.

We will use these to encourage $p_{ij}(z_i|z_j)$ to have high derivative magnitude (cliffs) at the same locations, independent of the values taken by $z_j$. This is done as follows. Let $\{\zeta_1, \ldots, \zeta_M\}$ be $M$ values of $z_j$, picked randomly from the training batch. We encourage the $\tilde{p}_{ij}(\cdot|z_j = \zeta_k)$ to be close to each other across the different $\zeta_k$ by minimizing their generalized Jensen-Shannon divergence (JSD) [2]:

$$\begin{aligned}
l_{\text{JSD}}(i|j) &= \text{JSD}\left[\tilde{p}_{ij}(\cdot|z_j = \zeta_1), \ldots, \tilde{p}_{ij}(\cdot|z_j = \zeta_M)\right] \\
&= \frac{1}{M}\sum_{k=1}^{m} D_{\text{KL}}(\tilde{p}_{ij}(\cdot|z_j = \zeta_k)\|\tilde{m}) \\
&= H(\tilde{m}) - \frac{1}{M}\sum_{k=1}^{M} H(\tilde{p}_{ij}(\cdot|z_j))
\end{aligned} \tag{14}$$

where $\tilde{m}(z_i) = \frac{1}{M}\sum_{k=1}^{M}\tilde{p}_{ij}(z_i|z_j = \zeta_k)$ and $D_{\text{KL}}$ denotes the Kullback-Leibler divergence. The differential entropy $H$ is defined as previously in Eq. 8.

This is repeated for each pair of variables $i, j$ from the $d$ variables, yielding the bivariate component of the loss:

---

[2]We have explored some divergences from the f-divergence family, such as the squared Hellinger distance, as well as other measures that are not in the f-divergence family, and found that JSD worked best in practice through the analysis described in Appendix C.

$$l_{\text{biv}} = \sum_{i=1}^{d} \sum_{j \neq i} l_{\text{JSD}}(i|j). \tag{15}$$

### 4.3 Preventing the collapse to Diracs

Lastly, we introduce a term to avoid degenerate solutions, such as the Dirac delta distribution mentioned earlier, or the collapse of the latent variables to a very small scale. It encourages the density of each standardized $z_i$ to be spread out not too unevenly, enforcing $p(z_i)$ for each dimension $i$ to be close to a uniform distribution. This is implemented as the KL divergence between these marginals and a uniform.

$$\begin{aligned} l_{\text{KL-uni}} &= \sum_{i=1}^{d} \text{KL}(U(-\sqrt{3}, \sqrt{3}), p(z_i)) \\ &= \sum_{i=1}^{d} (-2\sqrt{3} - \mathbb{E}_{z_i \sim U(-\sqrt{3}, \sqrt{3})}[\log(p(z_i))]) \end{aligned} \tag{16}$$

We use a uniform between $-\sqrt{3}$ and $\sqrt{3}$ because it has mean 0 and variance 1, as does the standardized $z_i$. The expectation is estimated as an average over $K$ samples from that uniform.

This term is useful even when the univariate criterion is not used, but only the bivariate. It encourages the support of the distribution to be learned more efficiently by "spreading" the distribution.

### 4.4 Total loss

We combine the three loss components defined into a weighted sum:

$$\mathcal{L}_{\text{Cliff}} = \lambda_{\text{uni}} l_{\text{uni}} + \lambda_{\text{biv}} l_{\text{biv}} + \lambda_{\text{KL-uni}} l_{\text{KL-uni}}, \tag{17}$$

where the corresponding $\lambda$ are hyperparameters controlling the relative strengths of each loss component. Training consists of learning the model parameters $\theta$ that produce the factors $z$ that minimize the loss $\mathcal{L}_{\text{Cliff}}$.

An interesting visualization of the loss landscape and how each term of the criterion promotes axis alignment is available in Appendix C.

**Computational complexity** We compute the loss for a batch of length $n$ with $d$ factors. We use $K$ values to estimate one-dimensional integrals along $z_i$, and $M$ different values for conditioning $z_j$.

The computational complexity is:

- univariate term $l_{\text{uni}}$: $O(dKn)$

- bivariate term $l_{\text{biv}}$: $O(d^2 MKn)$

- anticollapse term $l_{\text{KL-uni}}$: $O(dKn)$.

So the total loss computation has an overall complexity of $O(d^2 MKn)$.

## 5 Experiments

To assess if the proposed criterion for generic nonlinear transformations is effective for unsupervised disentanglement and prove its usefulness against current methods, we present three sets of experiments to answer the following questions:

1. Can Cliff correctly estimate latent variables under nonlinear transformations?

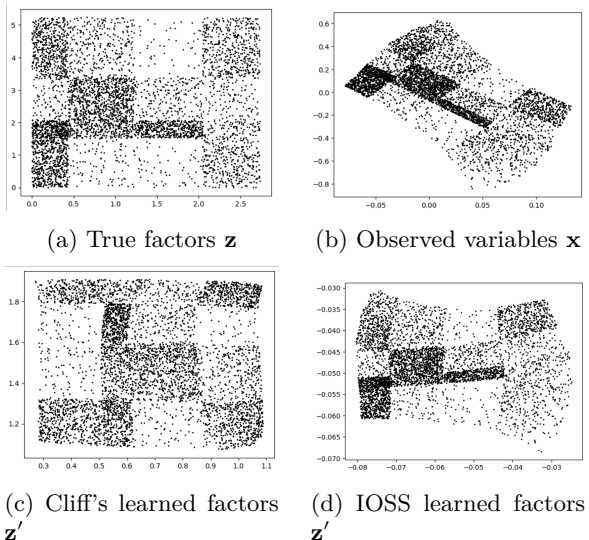

Figure 3: Synthetic data: True factors $z$ (a) are mapped through observed variables $x$ through a *nonlinear* mixing function. The nonlinearity is manifested through distortions. We learn a decoder $g$ that yields the reconstructed factor $z' = g(x)$. Our method, Cliff (c) matches the true factors (a) almost perfectly and obtains a much more straight and axis-aligned representation (MCC of $94.1 \pm 0.9$) than IOSS (d) (MCC of $91.6 \pm 0.8$) .

2. Does Cliff show a benefit when tested on a dataset that fulfills the assumption of axis-aligned discontinuities?

3. Is Cliff competitive against other disentanglement methods?

For question 1, we apply Cliff to a synthetic dataset with axis-aligned discontinuities, where it is possible to visualize the latent factors and their discontinuities, how they are deformed into observed space, how the reconstructed latent factors look, and whether they are aligned with the axes. Compared to other identifiable models, Cliff exhibits better alignment with the axes, which is the goal of disentanglement.

For question 2, we use a dataset with synthetically rendered images of balls (a variation of the dSprites dataset) whose latent variables (coordinates) have axis-aligned discontinuities. We observe that Cliff outperforms other models.

For question 3, we evaluate our method on the Shapes3D dataset, a widely used disentanglement benchmark, and compare it with other methods for unsupervised disentanglement, showing that our method demonstrates superior performance.

Regarding the main baselines, both the Independence-Of-Support Score (IOSS) (Wang & Jordan, 2021) and Hausdorff Factorized Support (HFS) (Roth et al., 2023) encourage the independence of the support of the latent factors, or equivalently, they both encourage the support to be factorized. However, they employ different estimation procedures that are appropriate depending on the dataset. Therefore, we will use IOSS for the first two questions and HFS for the last one. See Appendix D.5 for a more extensive discussion. All the experimental details, such as hyperparameter configurations and model architectures, are available in Appendix D.

## 5.1 Synthetic data

To answer question 1, we evaluate if Cliff can identify the true latent variables under nonlinear transformations in the simplest setting, with synthetic data and when the assumption is fulfilled. We use the synthetic dataset generation of the latent factors $z$ from Barin-Pacela et al. (2024), as it follows the axis-alignment

assumption. However, we replace the mixing function with a nonlinear mixing to visualize the distortion of the discontinuities in the observed variables. The dataset has axis-aligned discontinuities in $p_z$, as illustrated in Figure 3a. A nonlinear mixing function $f$ transforms $z$ onto $x$, as seen in Figure 3b.

Cliff's reconstruction $z'$ is very similar to $z$, as depicted in 3c, with clearly axis-aligned discontinuities and support. On the other hand, IOSS's (Wang & Jordan, 2021) reconstruction is not axis-aligned, as seen in Figure 3d.

For a quantitative comparison, we run both methods under 10 different initializations; Cliff reaches a Mean Correlation Coefficient (MCC) of 94.1±0.9, while IOSS reaches an MCC of 91.6±0.8. Therefore, for question 1, we conclude that not only Cliff can reconstruct latent variables in this simple and controlled nonlinear setting, but it also outperforms the current strongest baseline.

## 5.2 Balls dataset

To answer question 2, we test whether Cliff still succeeds in more realistic datasets of images, while the main assumption is still satisfied. We use a variant of the dSprites dataset (Matthey et al., 2017), the balls dataset from Ahuja et al. (2022a), where it is possible to control the distribution $p_z$ such that it has axis-aligned discontinuities. We develop the models and code from Lachapelle et al. (2023). We render two balls per image, each ball having its own color and whose coordinates are the latent variables that follow the same distribution as the latent factors of the synthetic data described in the previous section. That is, $z = (z_1, z_2, z_3, z_4)$, where $(z_1, z_2)$ are the coordinates of ball 1 and $(z_3, z_4)$ are the coordinates of ball 2. There are 4 axis-aligned discontinuities in $z_1$, 3 in $z_2$, 4 in $z_3$, and 3 in $z_4$.

We train an autoencoder with encoder $g_\phi$, decoder $f_\theta$, and a regularization term weighted by $\lambda_a$ accounting for the Cliff loss $\mathcal{L}_{\text{Cliff}}$ described previously. Thus, the optimized loss is

$$\mathcal{L} = \underbrace{\frac{1}{n}\|x - f_\theta(g_\phi(x))\|_2^2}_{\text{reconstruction error}} + \underbrace{\lambda_a \mathcal{L}_{\text{Cliff}}}_{\text{axis-alignment term}}. \tag{18}$$

We compare our method with the additive decoder (Lachapelle et al., 2022) and IOSS (Wang & Jordan, 2021) and evaluate them with the Mean Correlation Coefficient (MCC) with Spearman correlation coefficient since it gives the correlations up to nonlinear transformations. Similarly, the baselines are trained as an autoencoder with a regularization term encouraging the corresponding inductive bias, as established in the original evaluation for this task in Lachapelle et al. (2023). Both Cliff and IOSS are added as regularization terms on the autoencoder loss. We report the mean and standard error for each method. Table 2 shows that Cliff's MCC outperforms IOSS', whose assumption of factorized support also holds. The additive decoders perform the worst as there are overlaps between the balls. Therefore, for question 2, we conclude that Cliff achieves better results than both IOSS and Additive decoders, proving to be a suitable method when the model assumptions are fulfilled.

| Method | MCC |
|---|---|
| **Cliff** | **71.10 ± 2.98** |
| IOSS | 60.51 ± 5.58 |
| Additive Decoder | 37.80 ± 3.49 |

Table 2: Balls dataset Ahuja et al. (2022a): identification of the coordinates of two balls. Our model, Cliff, outperforms additive decoders (Lachapelle et al., 2022). Mean and standard error are reported for the MCC with the Spearman coefficient over 10 different initializations.

| Method | D | C | I | MIG |
|--------|---|---|---|-----|
| **Cliff** | **80.33 $\pm$ 2.60** | 68.52 $\pm$ 1.52 | 99.26 $\pm$ 0.30 | 28.54 $\pm$ 2.45 |
| HFS | 70.64 $\pm$ 4.77 | 78.29 $\pm$ 4.55 | 94.09 $\pm$ 2.07 | 58.82 $\pm$ 7.20 |
| $\beta$-VAE | 69.72 $\pm$ 3.54 | 63.84 $\pm$ 3.23 | 97.08 $\pm$ 1.43 | 24.60 $\pm$ 1.65 |

Table 3: Disentanglement scores for the Shapes3D dataset (Kim & Mnih, 2018). We report the Disentanglement (D), Completeness (C), Informativeness (I) (Eastwood & Williams, 2018), and Mutual Information Gap (MIG) (Chen et al., 2018). Our method (Cliff) outperforms Hausdorff Factorized Support (HFS) (Roth et al., 2023) and $\beta$-VAE (Higgins et al., 2017) on the disentanglement score D. Both Cliff and HFS are regularizers added to the $\beta$-VAE. The mean and its standard error are reported.

### 5.3 Disentanglement benchmarks

For question 3, we test the general usefulness of Cliff even when the assumption of axis-aligned discontinuities is not clearly fulfilled, with the goal being to determine if the overall method is useful for unsupervised disentanglement broadly and if it is competitive to the baselines. We evaluate our model on synthetically-rendered datasets that contain the true factors of generation of the images, such as pose, angle, color of the object, and background. This section focuses on the Shapes3D dataset (Kim & Mnih, 2018). This dataset contains 6 factors of variation: floor color, wall color, object color, object size, object type, and azimuthal angle of the object.

While the object type is a variable of discrete nature, all the others are of continuous nature. However, their values are a set of linearly spaced points, such that these variables can also be considered ordinal. Although Cliff considers continuous latent factors, its density estimation procedure relies on the Parzen window density estimator with bandwidth $\sigma$[3]. When a large enough $\sigma$ acts on a grid of ordinal values, the smoothed estimated density will have axis-aligned discontinuities satisfying our requirements.

We build on the "Disentangling Correlated Factors" library (Roth et al., 2023) and follow a similar methodology as established in the literature for this dataset (Roth et al., 2023; Locatello et al., 2019). Notably, 10 latent factors are estimated for all the methods presented. While being higher than the number of ground truth latent factors, this allows for considerable benefits in the optimization. We reserve for future work the study on the effect of the number of latent factors on optimization and identifiability. Yet, we observe that while overestimating the number of factors harms the compactness of the representation (and the completeness score will suffer), the representation will not necessarily be entangled since instead, multiple estimated factors may be redundantly representing the same one factor from the ground truth. This being said, we decide that for this task, the most relevant evaluation metric is the disentanglement score (D) from DCI (Eastwood & Williams, 2018), which is also in agreement with previous work (Roth et al., 2023).

We present results on the two main baselines: $\beta$-VAE (Higgins et al., 2017) and HFS (Roth et al., 2023). These baselines were chosen because the $\beta$-VAE is the simplest baseline on unsupervised disentanglement, and HFS is the main competitor of Cliff for this task. Both HFS and Cliff are implemented as a regularization term added to the $\beta$-VAE. For example, for Cliff, the loss on this task is given as

$$
\mathcal{L} = -\underbrace{\mathbb{E}_{z \sim q_\phi(z|x)} \log p_\theta(x|z)}_{\text{reconstruction term}} + \underbrace{\beta \mathrm{KL}(q_\phi(z|x) \,\|\, p_\theta(z))}_{\text{KL divergence term}}
$$
$$
+ \underbrace{\lambda_a \mathcal{L}_{\text{Cliff}}}_{\text{axis-alignment term}}
\tag{19}
$$

for a probabilistic encoder $q_\phi(z)$, decoder $p_\theta(x|z)$, and $\lambda_a$ being the regularization coefficient for the axis-alignment term.

For hyperparameter selection, we consider the initialization as one of the optimization hyperparameters to be chosen. Each set of hyperparameters is run for 10 seeds, and the best seed is selected based on the D

---

[3]For different datasets than the ones reported here, it may be necessary to do a hyperparameter search to find the best $\sigma$.

score. Then, the mean and standard error are computed for the selected set of hyperparameters for each of the scores. The details about the hyperparameter grid and the optimal hyperparameters selected are in Appendix D.

As seen in Table 3, Cliff obtains the best D score compared to both HFS and $\beta$-VAE. We notice that while the $C$ and $I$ scores are not high, this is to be expected since the theory guarantees only the *quantized* identification of factors. The discussion of the DCI-ES score and its connection to identifiability from Eastwood et al. (2023) implies that only when all of the D, C,I , E and S scores are close to 1 (or 100 in this scaled case), precise identifiability up to scale and permutation is possible. Without the E score (which is severely computationally expensive), it is difficult to make more conclusions about where exactly in the identifiability spectrum our model lies, but it seems to hint at possible indeterminacy inside the quantization, as expected from this criterion.

Therefore, for question 3, we conclude that Cliff is a useful method for unsupervised disentanglement even when its main assumption is not completely fulfilled (the discontinuities are not pronounced), demonstrating potential evidence for the practicality of non-global assumptions about the density discussed previously. The high disentanglement score of $80.33 \pm 2.60$ on this task is evidence of good identification of the latent variables, and beyond that, it also outperforms the baselines.

## 6 Conclusion

We tackle the problem of nonlinear unsupervised disentanglement and propose a criterion for aligning the discontinuities in the density of the learned latent factors with the axes. This criterion is based on the theory of identifiability of quantized factors, for which a criterion had been proposed only for the linear case. We extend current disentanglement benchmarks for a more reliable evaluation and show that our method, Cliff, outperforms the other methods according to the MCC and DCI scores.

Our empirical evaluation answers three important questions. First, the proposed criterion can better identify the latent variables than IOSS, and the axis alignment can be verified visually. Second, we verify Cliff's performance in a dataset where its assumptions of axis-aligned discontinuities are fulfilled. The improvement over the other methods showcases the benefits of applying this method in real datasets satisfying the axis-alignment assumption (which were motivated in Barin-Pacela et al. (2024)). Third, we demonstrate that Cliff is competitive against other methods for unsupervised disentanglement in the Shapes3D benchmark.

This work illustrates the potential of completely unsupervised disentanglement methods, a promising endeavor for real-world datasets. In future work, we hope to evaluate the usefulness of reusing disentangled representations learned through Cliff in an unsupervised manner for downstream tasks, hoping to improve sample efficiency and worst-group accuracy for example.

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

## A  Appendix

## B  Summary of theorems from "On the Identifiability of Quantized Factors"

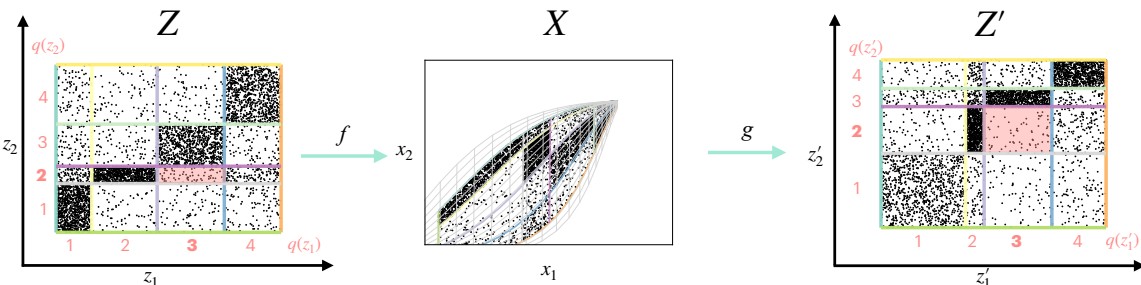

Figure 4: **Figure from Barin-Pacela et al. (2024). Recovery of quantized factors.** **Left:** The true (continuous) latent factors $Z_1$ and $Z_2$ are not independent, but their joint probability density $p_Z$ has *independent discontinuities*: sharp changes in the density that are aligned with the axes and form a grid. **Middle:** The factors get warped and entangled by the diffeomorphism $f$ into observations $X$, but the discontinuities in their density survive in the observed space. **Right:** We can learn a diffeomorphism $g$ that yields a density $p_{Z'}$ having axis-aligned discontinuities. This suffices to recover a grid whose cells match the initial grid's cells (up to possible permutation and axis reversal). **Pink cell example:** the points $Z'$ in cell $(3, 2)$ originated from the points $Z$ in cell $(3, 2)$. To construct these cells, the quantization of each continuous factor to an integer depends on thresholds based on the location of the discontinuities. The quantizations of $Z_1'$ and $Z_2'$ match precisely the quantizations of $Z_1$ and $Z_2$, up to possible permutation and axis reversal. This summarizes the *identifiability of quantized factors* under diffeomorphisms.

Here, we reintroduce the main definitions and theorems from Barin-Pacela et al. (2024). Figure 4 summarizes the main result from the main theorem (Theorem 2). For ease of reference, we reuse the figure and caption from Barin-Pacela et al. (2024).

### B.1  Main definitions and results

Setup (Barin-Pacela et al., 2024):

$$\underbrace{Z}_{\substack{\subset \mathbb{R}^d \\ \text{true factors}}} \sim p_Z \xrightarrow[\text{unknown}]{f} \underbrace{X}_{\substack{\subset \mathbb{R}^D \\ \text{observed data}}} \xrightarrow[\text{learned}]{g \approx f^{-1}} \underbrace{Z'}_{\substack{\subset \mathbb{R}^d \\ \text{recovered factors}}}$$

with $h = g \circ f$ spanning from $Z$ to $Z'$.

(**Barin-Pacela et al., 2024**) **Precise Identifiability of Factors:** Knowledge of $p_X$ is sufficient to determine a reverse mapping $g : \mathbb{R}^D \to \mathbb{R}^d$ that will yield recovered factors $(Z'_1, \ldots, Z'_d) = g(X)$ that correspond one-to-one to the ground-truth factors $(Z_1, \ldots, Z_d)$, up to permutation and component-wise invertible transformations (ideally monotonic).

(**Barin-Pacela et al., 2024**) **Identifiability of Quantized Factors:** Knowledge of $p_X$ is sufficient to determine a reverse mapping $g : \mathbb{R}^D \to \mathbb{R}^d$ that will yield recovered factors $(Z'_1, \ldots, Z'_d) = g(X)$ such that their quantization $(q'_1(Z'_1), \ldots, q'_d(Z'_d))$ will correspond one-to-one to the quantized ground-truth factors $(q_1(Z_1), \ldots, q_d(Z_d))$, up to possible permutation of indices and order reversal.

**Definition B.1.** (Barin-Pacela et al., 2024) Let $\mathcal{S}$ be the support of $p_Z$. We say that $p_Z$ has an **independent discontinuity** at $Z_i = \tau$ when every point in the intersection of the *coordinate hyperplane* $\{\mathbf{z}_i = \tau\}$ with $\mathcal{S}$ is a non-removable discontinuity of $p_Z$. Formally, this independent discontinuity at $Z_i = \tau$ is defined as the set $\Gamma_{\mathcal{S}}(i, \tau) = \{\mathbf{z} \in \mathcal{S} | \mathbf{z}_i = \tau\}$ under the condition that $\forall \mathbf{z} \in \Gamma_{\mathcal{S}}(i, \tau)$, $p_Z$ has a non-removable discontinuity at $\mathbf{z}$.

### B.1.1 Summary of the main result (Barin-Pacela et al., 2024)

**Assumptions**

- $f$ is a diffeomorphism

- $(Z_1, \ldots, Z_d) \sim p_Z$ are $d$ continuous random variables.

- The interior of the support of $p_Z$ is a connected set.

- The set of non-removable discontinuities of $p_Z$ is the union of a finite set of independent discontinuities that together form an *axis-aligned grid*. This grid must also possess a *backbone*.

**Quantized factor identifiability theorem** Under the above assumptions:

- It suffices to learn a diffeomorphism $g$ yielding $Z' = g(X)$ such that the PDF of $p_{Z'}$ has independent discontinuities forming an axis-aligned grid.

- Then, the quantized reconstructed factors $(q'_1(Z'_1), \ldots, q'_d(Z'_d))$ will correspond one-to-one to the quantized ground-truth factors $(q_1(Z_1), \ldots, q_d(Z_d))$, up to possible permutation of indices (and order reversal).

- The quantization thresholds used for $q_i$ and $q'_i$ are obtained as the locations of the independent discontinuities.

## B.2 Main theorems

**Theorem 1.** *(Barin-Pacela et al., 2024) Grid structure preservation and recovery theorem. Let $h : \mathcal{S} \subset \mathbb{R}^d \to \mathcal{S}' \subset \mathbb{R}^d$ be a diffeomorphism, where both $\mathcal{S}$ and $\mathcal{S}'$ are open connected subsets of $\mathbb{R}^d$. Suppose we have an axis-aligned grid $G \subset \mathcal{S}$, associated with its axis-separator-set $\mathcal{G}$ and discrete coordination $\mathbf{A}$, that is, $G = \mathrm{grid}_{\mathcal{S}}(\mathbf{A})$. While the grid does not need to be "complete", we suppose that $\mathcal{G}$ has at least one backbone. Now, suppose that we have another axis-aligned grid in $\mathcal{S}'$, associated with its discrete coordination $\mathbf{B}$, with $G' = \mathrm{grid}_{\mathcal{S}'}(\mathbf{B})$. Suppose $G' = h(G)$. Then, there exists a permutation function $\sigma$ over dimension indexes $1, \ldots, d$ and a direction reversal vector $s \in \{-1, +1\}^d$ such that $\forall j \in \{1, \ldots, d\}$, $i = \sigma^{-1}(j)$, $K = |\mathbf{A}_i| = |\mathbf{B}_j|$, $\forall k \in \{1, \ldots, K\}, \forall z' \in \mathcal{S}'$,*

*If $s_i = +1$, then:*

$$\begin{cases} z'_j = \mathbf{B}_{j,k} \Longleftrightarrow h^{-1}(z')_i = \mathbf{A}_{i,k}, \\ z'_j > \mathbf{B}_{j,k} \Longleftrightarrow h^{-1}(z')_i > \mathbf{A}_{i,k}, \\ z'_j < \mathbf{B}_{j,k} \Longleftrightarrow h^{-1}(z')_i < \mathbf{A}_{i,k}; \end{cases}$$

*If $s_i = -1$, then:*

$$\begin{cases} z'_j = \mathbf{B}_{j,k} \Longleftrightarrow h^{-1}(z')_i = \mathbf{A}_{i,K-k+1}, \\ z'_j > \mathbf{B}_{j,k} \Longleftrightarrow h^{-1}(z')_i < \mathbf{A}_{i,K-k+1}, \\ z'_j < \mathbf{B}_{j,k} \Longleftrightarrow h^{-1}(z')_i > \mathbf{A}_{i,K-k+1}. \end{cases}$$

**Corollary 1.** *([Barin-Pacela et al., 2024](#)) **Recovery of quantized factors.** Under the same premises as Theorem [1](#), consider random variables $Z$ and $Z' = h(Z)$. Using the quantization operation $Q$, we recover quantized factors up to permutation $\sigma$ of the axes and possible direction reversal indicated by $s$: $\forall i \in 1, \ldots, d$, $Q(Z_i; \mathbf{A}_i) = Q^{s_i}(Z'_j; \mathbf{B}_j)$ with $j = \sigma(i)$.*

**Theorem 2.** *([Barin-Pacela et al., 2024](#)) **Quantized factors identifiability theorem.** Let $Z$ be a latent random variable with values in $\mathcal{Z} \subset \mathbb{R}^d$ and whose PDF is $p_Z$. Let $f : \mathcal{Z} \to \mathcal{X} \subset \mathbb{R}^D$ be a diffeomorphism, and $X = f(Z)$ be the observed random variable. Assume that the support of the PDF $p_Z$ is an open connected set[4]. Further assume that $p_Z$ has at least one connected independent discontinuity in each dimension, such that the set of non-removable discontinuities of $p_Z$ forms an axis-aligned grid with a backbone. Let $\mathbf{A}$ be the discrete coordination of this grid. Then, there exists a diffeomorphism $g : \mathcal{X} \to \mathcal{Z}'$ yielding a variable $Z' = g(X)$ such that the set of non-removable discontinuities of the PDF $p_{Z'}$ is an axis-aligned grid. Consider any such diffeomorphism $g$, and let $\mathbf{B}$ be the discrete coordination of its resulting axis-aligned grid. Then, there exists a permutation function $\sigma$ over the dimension indexes $1, \ldots, d$, and a direction reversal vector $s \in \{-1, +1\}^d$ such that $q'_j(Z'_j) = q_i(Z_i)$ with $i = \sigma^{-1}(j)$, where $q'_j(Z'_j) = Q^{s_i}(Z'_j; \mathbf{B}_j)$ and $q_i(Z_i) = Q(Z_i; \mathbf{A}_i)$. In other words, the quantized factors in $Z'$ agree with the quantized factors in $Z$, up to permutation and possible axis reversal.*

### B.3 Criterion

Here, we further discuss how Cliff differs from the criterion from [Barin-Pacela et al. (2024)](#), the latter being empirically successful only when the mixing function is linear.

Their criterion is designed to align the output of linear maps with the axes, but it is not sufficient for nonlinear maps since they can completely distort the grid and it is important not only to align it with the axes but to straighten the mapping too. We have verified this empirically but did not include the comparison because we did not find it appropriate since it's not designed for nonlinear maps.

In short, their criterion estimates the joint density $\hat{p}_\sigma$ of $z$ and uses it to obtain the gradients $\frac{\partial \log \hat{p}_\sigma}{\partial z}$. Then, they encourage the alignment of these gradient vectors with the standard basis vectors (axes) by maximizing their cosine similarity.

In contrast, we estimate the joint density (and its respective gradients) only pairwise for two variables. We also estimate the marginal and conditional of the latent factors, which is more scalable in high dimensions and is the core of what is employed in each of our terms. Meanwhile, the criterion from [Barin-Pacela et al. (2024)](#) depends completely on the joint density between all the factors, the estimation of which may not be reliable on high dimensions. Finally, and most importantly, the balance of our three terms can straighten grids that are completely warped by diffeomorphisms.

## C  Validating the criterion

We would like to verify how each term of the criterion encourages the discontinuities to be aligned with the axes. For this, we can do a "grid search" over all the possible projections (in all possible directions), over the two latent variables: $\mathbf{z} = (z_1, z_2)$. We want to learn two vectors, $\mathbf{w}_1$ and $\mathbf{w}_2$, which should lead $z'_1$ and $z'_2$ to be axis-aligned. That is, we project $\mathbf{z}$ onto $\mathbf{w}_1$: $\text{proj}_{\mathbf{w}_1} \mathbf{z} = \frac{\mathbf{z} \cdot \mathbf{w}_1}{||\mathbf{w}_1||^2} = (||\mathbf{z}|| \cos \theta_1) \hat{\mathbf{w}}_1$, where $\theta_1$ is the angle between $\mathbf{z}$ and $\mathbf{w}_1$, and $\hat{\mathbf{w}}_1$ is the direction of the vector $\mathbf{w}_1$ with unit length.

Similarly, $\text{proj}_{\mathbf{w}_2} \mathbf{z} = \mathbf{z} \cdot \mathbf{w}_2 = \cos \theta_2 \hat{\mathbf{w}}_2$. For simplicity, we can consider that $\mathbf{z}$, $\mathbf{w}_1$, and $\mathbf{w}_2$ have unit norm.

---

[4]Alternatively, if the support is not open, we can consider its interior.

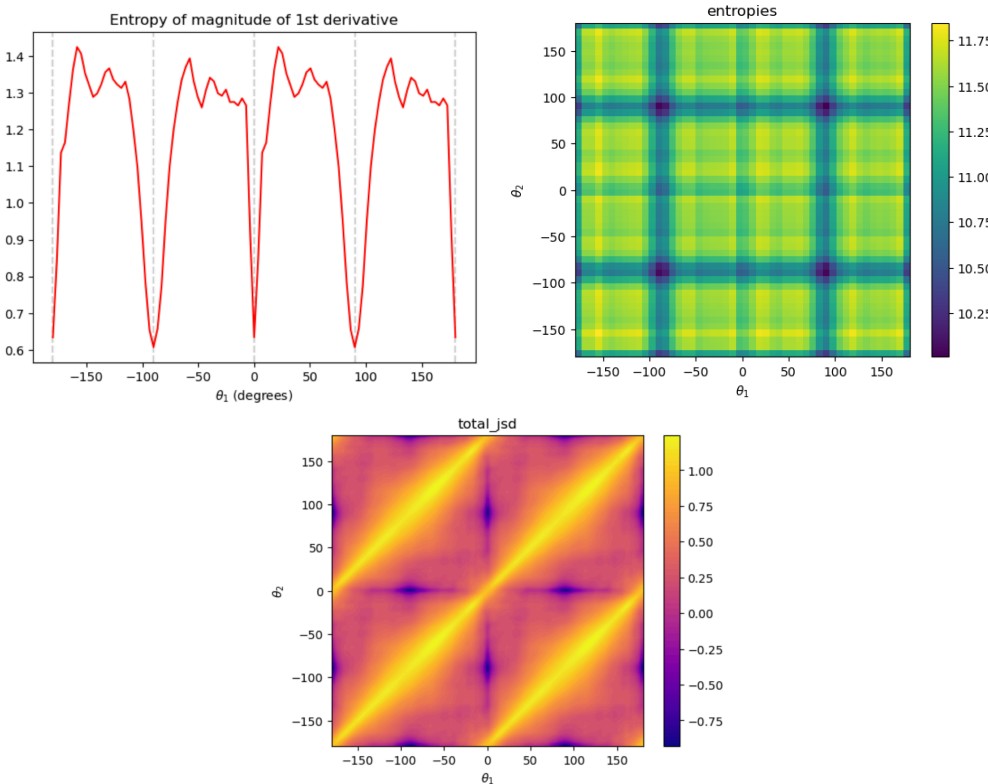

Figure 5: Loss landscape. Top row: Univariate criterion (encouraging cliffs in the marginals), minimized at 0° or 90°. Bottom row: Bivariate criterion (encouraging independent cliffs): avoids $\theta_1 = \theta_2 = 0°$ and $\theta_1 = \theta_2 = 90°$; instead, the minimum is at $(0°, 90°)$ and its multiples.

We design a "projection matrix"

$$\mathbf{W} = \left[ \begin{array}{c} \mathbf{w}_1^\mathsf{T} \\ \mathbf{w}_2^\mathsf{T} \end{array} \right]$$

such that $\mathbf{z}' = \mathbf{W}\mathbf{z}$.

More precisely, $\mathbf{w}_1 = (\cos\theta_1, \sin\theta_1)$, since $w_{1,1}$ is the coordinate of $\mathbf{w}_1$ in the $z_1$ axis, and $w_{1,2}$ is the coordinate of $\mathbf{w}_2$ in the $z_2$ axis. Similarly, $\mathbf{w}_2 = (\cos\theta_2, \sin\theta_2)$. With this, we can obtain all the possible parameterizations as a function of $\theta_1$ and $\theta_2$:

$$\mathbf{z}' = \mathbf{W}\mathbf{z} = \left[ \begin{array}{c} \mathbf{w}_1^\mathsf{T}\mathbf{z} \\ \mathbf{w}_2^\mathsf{T}\mathbf{z} \end{array} \right] = \left[ \begin{array}{c} z_1\cos\theta_1 + z_2\sin\theta_1 \\ z_1\cos\theta_2 + z_2\sin\theta_2 \end{array} \right] = \left[ \begin{array}{c} \mathrm{proj}_{\mathbf{w}_1}\mathbf{z} \\ \mathrm{proj}_{\mathbf{w}_2}\mathbf{z} \end{array} \right]. \tag{20}$$

This enables us to see the angles that minimize the loss. In Figure 5, the first term of the criterion (univariate – encouraging cliffs in the marginals) is represented in the top row. On the left, the loss is minimized for 0°, 90°, and their multiples. On the right, this same univariate loss is depicted for two angles at the same time. The second row presents the bivariate criterion (encouraging independent cliffs). The role of this term is to prevent $\theta_1 = \theta_2$ at the minimum. The minimum of this term is shown in purple as combinations of 0 and 90 degrees, but they are never the same (as opposed to the univariate criterion).

## C.1 Latent traversals example

The estimated latent factors can be visualized through the images in Figure 6, where each row corresponds to a particular factor, and each column corresponds to a traversal across this factor. The aim of disentanglement

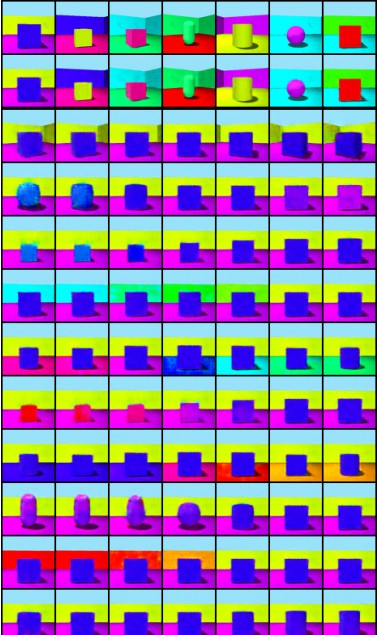

Figure 6: Shapes3D latent traversals.

is that each factor should be unique and should not affect the others. For example, in the third row, we observe that the only attribute that changes across different traversals (columns) is the angle, while all the others are kept fixed (object color, wall color, object size, object shape), which is exactly what we desire. On the other hand, the first row is changing various attributes across traversals, so this representation is not disentangled yet.

## D    Experimental details

All the experiments are executed with the Adam optimizer with default parameters (apart from the learning rate).

### D.1    Density estimation training

First, we train a Parzen window density estimator on the (standardized) joint samples to obtain $p_\sigma(z_1^{\text{train}}, z_2^{\text{train}})$. Then, we select a test set consisting of $z_1$ being 100 linearly spaced samples between -5 and 5, and $z_2$ being the first 20 samples of the test set. We evaluate the density estimator on this test set to obtain $p_\sigma(z_1^{\text{test}}, z_2^{\text{test}})$. We also evaluate the marginal $p_\sigma(z_2^{\text{test}})$, from which we can obtain the conditional $p_\sigma(z_1^{\text{test}}|z_2^{\text{test}}) = p_\sigma(z_1^{\text{test}}, z_2^{\text{test}}/p_\sigma(z_2^{\text{test}}))$.

### D.2    Synthetic dataset

We conducted an extensive empirical investigation and found that as long as the neural network (encoder g) has enough capacity, Adam always converges to the desired solution in the datasets with axis-aligned discontinuities in $p_z$.

Hyperparameters for the results reported:

- learning rate = 0.002

- batch size = 5000

- number of epochs = 1000

- $\lambda_{\text{uni}} = 0.0$

- $\lambda_{\text{biv}} = 1.0$

- $\lambda_{\text{KL-uni}} = 1.0$

- number of datasets $= 10$

- number of samples in the dataset $= 5000$

### D.3    Balls dataset

We follow the setting from Lachapelle et al. (2023) (including the encoder and decoder architectures):

- Dataset size: 20000

- Batch size: 64

- Learning rate: 0.001

- Number of seeds (initialization): 10

- number of epochs: 1000

For both our criterion and IOSS, we search over regularization strengths $\lambda_a \in \{0.1, 0.5, 1.0\}$. For Cliff, the result is reported for optimal $\lambda_a^* = 0.1$, and for IOSS, $\lambda_a^* = 1.0$.

Cliff-specific hyperparameters:

- Learning rate (lr): $10^{-5}$

- Number of epochs: $10^5$

- $\lambda_{\text{uni}}^* = 0.5$

- $\lambda_{\text{biv}}^* = 1.0$

- $\lambda_{\text{KL-uni}}^* = 0.7$

- $\sigma = 0.1$

We have searched over $\lambda_{\text{uni}} \in \{0.2, 0.5, 1.0\}$, $\lambda_{\text{KL-uni}} \in \{0.1, 0.5, 0.7\}$ lr $\in \{10^{-5}, 10^{-4}, 10^{-3}\}$.

**Additive Decoder:**    "An additive decoder has the form $f(z) = \sum_{B \in \mathcal{B}} f^{(B)} z_B$. Each $f(B)$ has the same architecture as the one presented above for the nonadditive case, but the input has dimensionality $|B|$" (Lachapelle et al., 2022).

### D.4    Shapes3D

We reuse the hyperparameters from Roth et al. (2023):

- learning rate $= 0.0001$

- batch size $= 64$

- number of epochs $= 100$

- evaluation batch size $= 1000$

- model architecture for VAE from Locatello et al. (2019)

- number of estimated factors = 10

- number of initialization (seeds) = 10

For searching over the hyperparameters of Cliff's criterion $\lambda_{\text{uni}}$, $\lambda_{\text{biv}}$, $\lambda_{\text{KL-uni}}$, $\lambda_a$, we perform a series of experiments to determine the grid range. When running the $\beta$-VAE, we observe that $\beta = 10$ gives a high D score for $\beta \in \{1, 2, 4, 8, 10, 16\}$, so we fix $\beta$ to 10 for all the experiments since there are already 4 other hyperparameters to search over. Initially, we take $\lambda_{\text{uni}} \in \{0.1, 0.2, 0.5, 0.7, 1.0\}$, $\lambda_{\text{biv}} = 10$, $\lambda_{\text{KL-uni}} \in \{0.1, 0.2, 0.5, 0.7, 1.0\}$, $\lambda_a = \{0.1, 0.5, 1.0, 10, 16, 100\}$, and $\sigma \in \{0.1, 0.2, 0.5\}$. Some of these combinations were discarded early in training for not being optimized easily.

For the baselines, we search over optimal hyperparameters as well. For the $\beta$-VAE, we search over $\beta$ in the range already mentioned, and for HFS, we search over both $\beta$ and $\gamma \in \{1, 10, 100\}$.

The following hyperparameters were found to be optimal and are used in the results reported:

- Cliff: $\lambda_{\text{uni}} = 0.5$, $\lambda_{\text{biv}} = 1.0$, $\lambda_{\text{KL-uni}} = 0.7$, $\lambda_a = 1$, $\sigma = 0.1$.

- HFS: $\beta = 1$, $\gamma = 100$.

- Beta VAE: $\beta = 16$.

### D.5 Discussion on IOSS vs HFS

Both IOSS and HFS enforce the same inductive bias of factorized support, although there are slight differences in the implementation.

The choice of one over the other is merely practical due to how much work it takes to tune the model for the particular dataset, since we wish the baseline to be as strong as possible. HFS has already been trained on Shapes3D by Roth et al. and we reuse their implementation to guarantee the best results since it relies on particular estimation procedures and setups. For the balls dataset, neither IOSS nor HFS has been tuned as a baseline, but IOSS is significantly easier to train due to its simple, modular and compact implementation, as well as the stability of the method, which is why we chose it for the other datasets.

Expanding on the differences, first, we notice that "Independence Of Support" and "Factorized Support" are equivalent notions, both defined by:

$$\text{supp}\,(Z_1, \ldots, Z_d) = \text{supp}\,(Z_1) \times \cdots \times \text{supp}\,(Z_d) \tag{21}$$

This is computed through the Hausdorff distance between the set of mapped datapoints and a set of points with factorized support. The HFS loss employs the Monte Carlo Hausdorff distance estimation and further relaxing of the factorized support into pairwise factorization. The autoencoder term is crucial to avoid collapse and retain input information. Therefore, we highlight that the HFS regularization term cannot be employed on its own, and hyperparameter optimization always needs to happen together with the autoencoder terms, which makes it difficult to use in practice, as well as model-dependent, while our proposed criterion is model agnostic.

Interestingly, IOSS is very similar to what we are proposing with $l_{KL-uni}$ in our criterion. In fact, we also have an implementation of this criterion in the bivariate case, where we draw samples from 2D uniform distributions, and the result should also have factorized support. Therefore, we conclude that our criterion also encourages factorized support.

## E  Model architectures

### E.1  Architectures for synthetic dataset

Encoder:

- Linear layer (2, 50) followed by tanh

- Linear layer (50, 100) followed by tanh

- Linear layer (100, 50) followed by tanh

- Linear layer (50, 2).

Ground-truth decoder: x = B tanh A (0.5 z)

### E.2  Architectures from Lachapelle et al. (for balls dataset)

**Encoder**:

- RestNet-18 Architecture till the penultimate layer (512 dimensional feature output)

- Stack of 5 fully-connected layer blocks, with each block consisting of Linear Layer ( dimensions: 512 × 512), Batch Normalization layer, and Leaky ReLU activation (negative slope: 0.01).

- Final Linear Layer (dimension: 512 × d) followed by Batch Normalization Layer to output the latent representation.

**Decoder** (Non-additive):

- Fully connected layer block with input as latent representation, consisting of Linear Layer (dimension: $d_z$ × 512), Batch Normalization layer, and Leaky ReLU activation (negative slope: 0.01).

- Stack of 5 fully-connected layer blocks, with each block consisting of Linear Layer ( dimensions: 512 × 512), Batch Normalization layer, and Leaky ReLU activation (negative slope: 0.01).

- Series of DeConvolutional layers, where each DeConvolutional layer is follwed by Leaky ReLU (negative slope: 0.01) activation.

    - DeConvolution Layer (cin: 64, cout: 64, kernel: 4; stride: 2; padding: 1)
    - DeConvolution Layer (cin: 64, cout: 32, kernel: 4; stride: 2; padding: 1)
    - DeConvolution Layer (cin: 32, cout: 32, kernel: 4; stride: 2; padding: 1)
    - DeConvolution Layer (cin: 32, cout: 3, kernel: 4; stride: 2; padding: 1)

### E.3  Architectures from Locatello et al. (for Shapes3D dataset)

Encoder:

Input: 64 × 64× number of channels

4 × 4 conv, 32 ReLU, stride 2

4 × 4 conv, 32 ReLU, stride 2

4 × 4 conv, 64 ReLU, stride 2

4 × 4 conv, 64 ReLU, stride 2

FC 256, F2 2 × 10

(Bernoulli) Decoder:

Input: $\mathbb{R}^{10}$

FC, 256 ReLU

FC, 4 × 4 × 64 ReLU

$4 \times 4$ upconv, 64 ReLU, stride 2

$4 \times 4$ upconv, 32 ReLU, stride 2

$4 \times 4$ upconv, 32 ReLU, stride 2

$4 \times 4$ upconv, number of channels, stride 2

