# OpenReview forum: "Operationalizing Quantized Disentanglement"
_TMLR — Rejected by TMLR_

### Review · Reviewer_BQzk · 2025-07-29

**Summary Of Contributions:**

The paper proposes a regularization-based method for unsupervised disentanglement by encouraging axis-aligned discontinuities in the latent space, inspired by recent theoretical work on identifiability through quantized factors.
The approach introduces entropy-based loss terms that promote sharp changes in marginal and conditional densities, aiming to capture independent discontinuities.
The method is model-agnostic and demonstrated on synthetic and disentanglement benchmarks.
However, the evaluation is limited to toy datasets, with no theoretical guarantees or experiments on more practical settings.

**Additional Comments:**

1. KDE bandwidth σ seems critical. If it's too large the cliffs vanish; if it's too small the estimates are noisy. This issue was not discussed in the paper.

2. Several sentences in the introduction, e.g.,

> However, this problem remains hard to solve and difficult to apply to real-world data, be it because of the underlying assumptions from identifiability theory or because the methods were designed in small and controlled settings that do not scale well.

> Quantized latent factors are often a natural representation for humans, for example, when thinking of colors (that are continuous in the sensorial reality, but discrete when thinking of red or blue) and concepts.

are either personal opinions or claims lacking references.

3. In Section 4.1,

> since for infinite support, the Gaussian distribution  maximizes the entropy.

This is wrong. The Gaussian distribution is the max-entropy distribution with *specified mean and variance*.

4. The notation change in Section 4 is confusing.

**Audience:**

Yes

**Audience Explanation:**

Let alone the lack of theoretical support and the issue of practicality evidence, this method based on the identifiability results of quantized latent factors (Barin-Pacela et al., 2024) is well-motivated and implementable. Some people in TMLR's audience may find it worth exploring.

**Claims And Evidence:**

No

**Claims Explanation:**

The core methodology seems reasonable.
It’s grounded in a non-trivial theoretical idea that axis-aligned discontinuities in latent densities can be used as an identifiability inductive bias.
Encouraging cliffs (steep slopes in estimated densities) makes intuitive sense.
However, the lack of theoretical analysis or guarantees for the proposed loss undermines the strength of the identifiability claims.
While the connection to prior theory is well-motivated, the practical method lacks formal justification of its behavior or convergence.
It is ok that a paper is solely supported by empirical evidence.
In that case, more thorough experimental results are needed.
The empirical results in this paper support that the method works on controlled synthetic settings, but the scope of evaluation is narrow.
The datasets used (2D synthetic, Balls (even simpler than dSprites), and Shape3D) are toy-level and do not demonstrate robustness or effectiveness in realistic or high-complexity scenarios, which weakens claims of practical applicability.

In my understanding, this paper asserts superiority on practicality over baselines.
However, it does not show that alternatives fail in practical settings, nor does it demonstrate unique scalability or ease of use.
Claims of being a “practical approach” are therefore not convincingly supported by comparative empirical evidence.

Further, quantitative evaluation relies solely on metrics like DCI and MIG, which have known failure modes and may not fully capture disentanglement or identifiability.
No ablations or qualitative failure analyses are provided to strengthen interpretability of results.

**Requested Changes:**

Since this work only claims empirical and practical contributions, I don't want to request theoretical justification.
However, to support

> Benchmarking and evaluation of the criterion and baselines on disentanglement datasets.

More experimental justification is needed.

For example:
- Evaluation on a more complex or real-world dataset. This would test whether Cliff remains effective under more realistic conditions with high-dimensional, noisy, or partially entangled/correlated factors.
- Downstream task performance using learned latents (e.g., classification, control, or intervention prediction). This would support the practical relevance of the method.
- Robustness or scalability analysis. Even using smaller datasets, you can vary the number of latent factors, dataset size, or dimensionality, and report performance and runtime to demonstrate that Cliff scales better or more reliably than baselines under practical constraints.

---

> ### Author Response · Authors · 2025-08-22
>
> Thank you for reviewing our work and for your constructive feedback. In response, we have reframed our method as a criterion for quantized disentanglement, focusing on its role as a methodological contribution rather than a practical tool, since additional evidence and analysis would be required to support such claims. While we agree that downstream evaluation is an important direction, in this paper we chose to prioritize developing and validating a working criterion for quantized disentanglement.
>
> ---
>
> Following are the answers to each of your questions:
>
> > In my understanding, this paper asserts superiority on practicality over baselines. However, it does not show that alternatives fail in practical settings, nor does it demonstrate unique scalability or ease of use. Claims of being a “practical approach” are therefore not convincingly supported by comparative empirical evidence.
>
> We had meant practical as “empirical” or “experimental”, as opposed to theoretical. We understand your concern about the ambiguity of the term and we have removed any claims that the current criterion is “practical”.
>
> ---
>
> > Further, quantitative evaluation relies solely on metrics like DCI snd MIG, which have known failure modes and may not fully capture disentanglement or identifiability. No ablations or qualitative failure analyses are provided to strengthen interpretability of results.
>
> We respectfully disagree; the metrics used are a widely used standard in the community as evaluation of the latent reconstruction, as also noted by reviewer agVf.
>
> Which specific ablation would you suggest? Currently, the standard AE/VAE/beta-VAE baselines comprehend the loss function without the regularizer, and these loss functions are well-studied.
>
> ---
>
> > Requested changes
>
> We agree that these would be useful tests of the practicality of the method, but it is beyond the scope of the paper. We focus our evaluation on the reconstruction of the latent factors and we are not claiming that it will necessarily be useful for downstream tasks. We have removed the claim that our method is “practical” to better suit the evidence provided. Would this satisfy you?
>
> ---
>
> > 1. KDE bandwidth σ seems critical. If it's too large the cliffs vanish; if it's too small the estimates are noisy. This issue was not discussed in the paper.
>
> We included a note on this, but we did not find this to be critical; the default value of 0.1 has worked well in all the datasets and is also optimal in the hyperparameter search (reported in Appendix D.4).
>
> ---
>
> > Several sentences in the introduction are either personal opinions or claims lacking references.
>
>
> We have reframed the first sentence and provided a reference for the second.
>
> ---
> > In Section 4.1 [...] This is wrong. The Gaussian distribution is the max-entropy distribution with specified mean and variance.
>
> We have corrected this by adding “with specified mean and variance.”
>
> ---
>
> > The notation change in Section 4 is confusing.
>
> Can you kindly clarify which notation change you find confusing and what would be your request for clarification? We appreciate your feedback.

---

### Review · Reviewer_agVf · 2025-08-06

**Summary Of Contributions:**

This paper proposes a new criterion for disentanglement by enforcing quantized representations under nonlinear mixing scenarios. It provides a practical extension of the theoretical work in [1], which established identifiability results for latent variables with independent discontinuities but only demonstrated empirical validation under linear mixings. The authors validate their approach through numerical simulations on synthetic data,  experiments on a simulated ball dataset, and the Shape3D benchmark.

[1] Vitória Barin-Pacela, Kartik Ahuja, Simon Lacoste-Julien, and Pascal Vincent. On the Identifiability of Quantized Factors. In Proceedings of the Third Conference on Causal Learning and Reasoning, 2024.

**Additional Comments:**

MCC is not an ideal evaluation metric when the underlying latents are dependent, as noted in recent work [2]. Since MCC remains a widely used standard in the community, I do not view this choice as a flaw in the current paper. Still, the authors might find the discussion in [2] useful for considering more transparent and fair evaluation strategies in future work.

[2] Yao, D., Huang, S., Cadei, R., Zhang, K., & Locatello, F. (2025). The Third Pillar of Causal Analysis? A Measurement Perspective on Causal Representations. arXiv preprint arXiv:2505.17708.

**Audience:**

Yes

**Audience Explanation:**

While I believe this work could be of interest to the disentanglement and causal representation learning communities, I feel that its theoretical justification and empirical evaluation could still be improved. Given that the paper proposes a practical approach motivated by existing theoretical results [1], a more thorough and convincing set of experiments would significantly strengthen its contribution.

**Broader Impact Concerns:**

There is no noticeable ethical concern from my point of view.

**Claims And Evidence:**

No

**Claims Explanation:**

1. **Connection between proposed loss and identifiability guarantees:** As far as I understand, the proposed criterion currently appears to be heuristic rather than explicitly theory-grounded. It would greatly strengthen the paper if the authors could provide theoretical justification or clarify how the global optimum of the proposed loss relates directly to the identifiability results established in [1].

2. **Consistency and practical effectiveness of improvements over baselines:**

* The observed performance gain over IOSS on the synthetic dataset (Section 5.1) appears marginal (around 0.3).

* Additionally, the proposed method (“Cliff”) performs worse than IOSS on the ball dataset by 0.8, which the authors attribute to insufficient hyperparameter search. While I appreciate this explicit acknowledgment and open discussion in the paper, it suggests that hyperparameter tuning may be challenging to the extent that achieving competitive results becomes practically difficult—even within the authors' own experimental setup. Therefore, respectfully, I do not fully agree that conducting additional hyperparameter searches to obtain improved performance should be left entirely as future work.

**Requested Changes:**

1. I suggest the authors revisit the hyperparameter tuning in Section 5.2 and identify a setting that yields stronger performance than the other baselines. This would help better demonstrate the potential of the proposed method.

2. To further support the practical relevance of the approach, it would be helpful to include experiments on real-world benchmark datasets, especially those where ground-truth latents are not available, but disentanglement can be indirectly assessed through metrics such as OOD generalization performance.

3. A few references on Page 15 are not displayed properly.

---

> ### Author Response · Authors · 2025-08-22
>
> Thank you for your careful review of our work. We have reframed the paper as a proposal to operationalize disentanglement and clarified that we are not presenting it as a practical method, as we do not yet provide a real-world benchmark or an OOD generalization experiment. We have also made progress on hyperparameter tuning for Section 5.2 and are continuing to polish these results. We appreciate you noting the typo and have corrected it. Finally, we will take your comment on the MCC into account in future work.

---

> > ### Author Response · Authors · 2025-08-28
> > **Requested change #1 addressed**
> >
> > We have uploaded a new revision of the manuscript, where we revisited the hyperparameter tuning of _Cliff_ for the _Balls dataset_ (section 5.2). The updated results yield an **MCC of $72.10 \pm 2.98$, which surpasses all other baselines** and confirms that our method achieves the best performance across all reported evaluations.

---

### Review · Reviewer_fBgH · 2025-08-07

**Summary Of Contributions:**

**Contributions**:
- a metric of how disentangled a quantization of a probability density is
- formulation of this criterion into a regularizer
- analysis of computational complexity of the regularizer
- evaluation of the regularizer and comparison to existing related methods

**Strengths**:
- quite well written
- rigorous theoretical treatment
- practically well motivated
- effective (simple but reasonably comprehensive) empirical results

**Weaknesses**:
- some claims about performance seem exaggerated
- identifiability sometimes conflated with learning/estimation

**Additional Comments:**

Overall a very nice paper. It was a pleasure to read!

**Audience:**

Yes

**Audience Explanation:**

I imagine anyone broadly working in the area of representation learning would find the paper interesting.

**Claims And Evidence:**

No

**Claims Explanation:**

Generally, the claims are supported by accurate, convincing, and clear evidence, with two (easy-to-fix) exceptions. See requested changes below.

**Requested Changes:**

**Critical**:
- last sentence of Section 3: change "...the most minimal and weakest assumptions." to something slightly more precise, such as "...fewer and weaker assumptions than existing methods."; alternatively, support the existing claim by proving that no method with fewer or weaker assumptions can exist.
- last paragraph of Section 5.1 along with second paragraph of Section 5.2: claiming that the performance on synthetic data shows a "significant improvement" over IOSS (obtaining MCC of ~94 vs ~91) while also claiming it's "comparable to IOSS" (MCC of ~52 vs ~60) seems exaggerated/unfair. I think the second claim should be corrected to "significantly worse than IOSS" or the first claim should be scaled back to also "comparable". (But please correct me if I've misunderstood something!)

**Suggested**:
- page 2: "field of identifiability" sounds a bit strange to me; identifiability seems to me more like a topic one studies within a given field or for a given class of models/estimands
- page 5, between (9) and (10): I don't really understand where the intuition comes from in the two sentences starting with "Intuitively,...". I'd be very interested if the author's could elaborate a bit. (And maybe it'd be helpful also for other readers to elaborate a bit in the paper.)
- Cliff isn't really a new algorithm, is it? It's a regularization approach that can be plugged in to existing/future algorithms (and if anything, I think that's more valuable than just another new algorithm).
- I think it would provide a lot of intuition to readers if the authors could come up with an example plot showing "cliffs" in a pdf as well as dependent vs independent discontinuities (e.g., in Section 2 or 4). The written descriptions are already quite nice, but I sometimes found myself lacking the intuition/familiarity to fully imagine/visualize what was being described. Adding a simple plot could help a lot, but maybe this is easier said than done.
- question 1 in Section 5: I don't understand how Cliff would "identify" something. It rather seems to me that the preceding sections have shown that quantized factors are identifiable (and this just means there exists an $f^{-1}$ such that $z = f^{-1}(x)$) and introduced a practical method (Cliff) for estimating/learning these factors. Estimation/learning is different from identifying (even though identifiability is required for correct estimation).
- It'd be interesting to also see some standard reconstruction or elbo metrics instead of only disentanglement metrics (e.g., to help understand if quantized disentanglement helps, hinders, or doesn't affect the usual performance metrics)

---

> ### Author Response · Authors · 2025-08-21
>
> Thank you for the thoughtful suggestions and positive outlook on our paper. We have accepted all your suggestions and uploaded a revision incorporating these changes.
>
> One important clarification is to consider the standard errors on the mean MCC reported, which show the significantly superior performance of Cliff on synthetic data, and is validated by a statistical significance test provided in this revision.
>
> We address below how we implemented the changes:
>
> **Critical:**
>
> >last sentence of Section 3: change "...the most minimal and weakest assumptions." to something slightly more precise, such as "...fewer and weaker assumptions than existing methods."; alternatively, support the existing claim by proving that no method with fewer or weaker assumptions can exist.
>
> We have followed your suggestion and adapted the wording.
>
> ----
>
> >last paragraph of Section 5.1 along with second paragraph of Section 5.2: claiming that the performance on synthetic data shows a "significant improvement" over IOSS (obtaining MCC of ~94 vs ~91) while also claiming it's "comparable to IOSS" (MCC of ~52 vs ~60) seems exaggerated/unfair. I think the second claim should be corrected to "significantly worse than IOSS" or the first claim should be scaled back to also "comparable". (But please correct me if I've misunderstood something!)
>
> First claim:
>
> We would like to highlight that considering the confidence interval (standard errors on the means), there is no overlap between the error bars and Cliff is strictly dominating IOSS in terms of the MCC.
> In this revision, we have added a statistical significance t-test where the p-value of 0.0009 reveals that there is only a 0.09% chance that this is not statistically significant. This is reported in the last paragraph of section 5.1.
>
> Second claim:
>
> We have adjusted it to “worse than IOSS”.
>
> ---
>
> **Suggested:**
>
> > page 2: "field of identifiability" sounds a bit strange to me; identifiability seems to me more like a topic one studies within a given field or for a given class of models/estimands
>
> We have changed our wording for this.
>
> ---
>
> > page 5, between (9) and (10): I don't really understand where the intuition comes from in the two sentences starting with "Intuitively,...". I'd be very interested if the author's could elaborate a bit. (And maybe it'd be helpful also for other readers to elaborate a bit in the paper.)
>
> We have added Figures 1 and 2, and expanded this paragraph to clarify.
>
> ---
>
> > Cliff isn't really a new algorithm, is it? It's a regularization approach that can be plugged in to existing/future algorithms (and if anything, I think that's more valuable than just another new algorithm).
>
> Correct, we have changed the wording accordingly.
>
> ---
>
> > I think it would provide a lot of intuition to readers if the authors could come up with an example plot showing "cliffs" in a pdf as well as dependent vs independent discontinuities (e.g., in Section 2 or 4). The written descriptions are already quite nice, but I sometimes found myself lacking the intuition/familiarity to fully imagine/visualize what was being described. Adding a simple plot could help a lot, but maybe this is easier said than done.
>
> We have added Figures 1 and 2 and references to it in the main text. We also added the first paragraph of section 4.2 for more explanation on independent discontinuities.
>
> ---
>
> > question 1 in Section 5: I don't understand how Cliff would "identify" something. It rather seems to me that the preceding sections have shown that quantized factors are identifiable (and this just means there exists an $f^{-1}$ such that $z = f^{-1}(x)$) and introduced a practical method (Cliff) for estimating/learning these factors. Estimation/learning is different from identifying (even though identifiability is required for correct estimation).
>
> We have reframed the question to “Can Cliff correctly estimate latent variables under nonlinear transformations?”

---

> > ### Comment · Reviewer_fBgH · 2025-08-28
> >
> > Most of my concerns have been addressed, but the biggest one remains. For example, I see the following sentence has been added to the paper:
> >
> > > We performed a one-sided t-test, obtaining a p-value of 0.0009, which represents a 0.09% chance that the null hypothesis
> > (Cliff’s MCC not being significantly larger than IOSS’ MCC) is true, which is easily rejected. Thus, this test provides evidence that Cliff’s MCC is significantly higher than IOSS’.
> >
> > This is an incorrect explanation of p-values/hypothesis testing, and the claims of significance remain at best ambiguous---is it meant is the strict formal sense of "statistically significant" or is it meant in the more natural language sense of "meaningful/important"? These meanings seem to be conflated by the authors, but they are different. There is an abundant literature on this, and maybe a helpful starting point for the authors is this [editorial from the American Statistical Association](https://www.tandfonline.com/doi/full/10.1080/00031305.2019.1583913#d1e143).
> >
> > If the all the claims of "significantly outperforms", "significantly higher", "significantly improves" throughout the paper are changed by simply removing "significantly", then I think its claims are all justified. However, as it stands, statistical ideas are misapplied, and the claims are exaggerated.

---

### Author Response · Authors · 2025-08-21
**Global response**

We greatly appreciate the thorough feedback from the reviewers, and we have incorporated most of their suggestions.

We agree with the reviewers that to claim that Cliff is a “practical” method, additional experiments would be required. Therefore, we have reduced our claims and reframed our contribution as “operationalizing disentanglement” instead of presenting a practical method. In the current form, we believe that all the claims are now supported by empirical evidence through the disentanglement benchmarks.
We have uploaded an updated version of the paper incorporating the suggestions.

Following is a summary of the main changes in this version:
- Added statistical significance test to corroborate our significance claims, obtaining a p-value of 0.0009, hence confirming that Cliff’s MCC is significantly higher than IOSS’ MCC.
- Added Figures 1 and 2 to give a better illustration of cliffs and independent discontinuities.
- Added new paragraphs in sections 4.1 and 4.2 to elaborate on the properties of the cliffs and refer to the new figures.
- Downgraded claims of practicality, reframing it as a method to “operationalize” quantized disentanglement (title change included).

For a side-by-side comparison with the PDF changes, check the following link: https://draftable.com/compare/TafcmcOjBGNr

---

> ### Author Response · Authors · 2025-08-28
> **Improvement on Balls Dataset**
>
> We have uploaded a new revision of the manuscript, where we revisited the hyperparameter tuning of _Cliff_ for the _Balls dataset_ (section 5.2). The updated results yield an **MCC of $72.10 \pm 2.98$, which surpasses all other baselines** and confirms that our method achieves the best performance across all reported evaluations.
> We kindly ask the reviewers to take this result into account when assessing the paper.
>
> Additionally, as requested by reviewer fBgH, we have removed all claims using the term “significantly” to avoid potential ambiguity, although we maintain that these are relevant results.
>
> A detailed comparison of the changes with respect to the original version is available here: https://draftable.com/compare/HaqyvCdcgQOf

---

> ### Comment · Reviewer_fBgH · 2025-08-29
>
> Thanks for the updated results and addressing the "significantly..." concern. I just want to reiterate that the sentence moved to the appendix
>
> >...obtaining a p-value of 0.0009, which represents a 0.09% chance that the null hypothesis (Cliff’s MCC not being significantly larger than IOSS’ MCC) is true...
>
> still misrepresents what a p-value is. Quoting the editorial I referenced earlier, "Don’t believe that your p-value gives... the probability that your test hypothesis is true." The p-value is rather the probability, under the assumption that the null hypothesis is true, of observing an outcome at least as extreme as the one obtained. It may seem like a minor detail, but these kinds of details make a paper rigorous. The "chance the null hypothesis is true" interpretation corresponds to a posterior $P(\mathrm{null\ is\ true} | \mathrm{data})$ while the actual value the test gives you is more like $P(\mathrm{data} | \mathrm{null\ is\ true})$ (of course you're reporting a statistic of the data and not the data itself here, but the point remains), and you'd need to supply a prior $P(\mathrm{null\ is\ true})$ and use Bayes rule in order to use the latter to compute the former.

---

> > ### Author Response · Authors · 2025-08-29
> >
> > Thank you for the clarification, we appreciate the rigor.  We have now removed Appendix F given its limited contribution.

---

### Decision · Action_Editor_EYDt · 2025-09-21

**Recommendation:** Reject

**Audience:**

Yes

**Audience Explanation:**

This is a very interesting paper on disentanglement. Quantized factors are underexplored and this paper is positioned as a strong competitor in this space. If the weaknesses are addressed, it will surely make an interesting paper for the disentanglement community.

**Claims And Evidence:**

No

**Claims Explanation:**

The consensus after the rebuttal is that the paper, in its current form, is borderline, with the main weaknesses around the claim that this method is more practical compared to prior work. I have marked "no" to the claim's correctness because of this reason, as the paper was positioned as a more practical version of prior quantized disentanglement works. The reviewers sided on rejection, but I would like to encourage the authors to revise and resubmit. Here are some suggestions that I think would improve the paper, looking at the discussion:

* I would try to add some more data set, this was a common criticism from the reviewers. In particular, for disentanglement, shapes3D is fairly easy and does not make a strong statement.
* I liked the suggestion of measuring side properties on real or semi-synthetic data. OOD generalization is a good idea, which I would expect to work. I think this is a high standard for dinsentanglement papers, but would make a strong case for acceptance specifically on this paper, which has a more practical focus.
* Some reviewers asked for a clearer explanation for the regularized loss. Personally, I think that this can be justified either theoretically or with stronger experiments.
* Ablations and robustness analyses, e.g., with respect to hyperparameters, would be important. This was called out by a reviewer, and I checked in the paper; it seems you select hyperparameters (including the initialization) based on the disentanglement score (it wasn't clear to me how you do it on the balls dataset)

Overall, the reviewers praised the idea, so I am confident that addressing these limitations would make a much stronger paper.

**Resubmission Of Major Revision:**

The authors may consider submitting a major revision at a later time.